# Candidate Genes and Pathways Associated with Gilles de la Tourette Syndrome—Where Are We?

**DOI:** 10.3390/genes12091321

**Published:** 2021-08-26

**Authors:** Amanda M. Levy, Peristera Paschou, Zeynep Tümer

**Affiliations:** 1Kennedy Center, Department of Clinical Genetics, Copenhagen University Hospital, Rigshospitalet, 2600 Glostrup, Denmark; marie.amanda.bust.levy@regionh.dk; 2Department of Molecular Biology and Genetics, Democritus University of Thrace, 68100 Alexandroupolis, Greece; ppaschou@gmail.com; 3Department of Biological Sciences, Purdue University, West Lafayette, IN 47907, USA; 4Department of Clinical Medicine, Faculty of Health and Medical Sciences, University of Copenhagen, 2200 Copenhagen, Denmark

**Keywords:** Gilles de la Tourette syndrome, GTS, tics, human genetics, neurotransmission, neurodevelopmental disorders, genome-wide association study, genetic association, cross-disorder, diagnostics

## Abstract

Gilles de la Tourette syndrome (GTS) is a childhood-onset neurodevelopmental and -psychiatric tic-disorder of complex etiology which is often comorbid with obsessive-compulsive disorder (OCD) and/or attention deficit hyperactivity disorder (ADHD). Twin and family studies of GTS individuals have shown a high level of heritability suggesting, that genetic risk factors play an important role in disease etiology. However, the identification of major GTS susceptibility genes has been challenging, presumably due to the complex interplay between several genetic factors and environmental influences, low penetrance of each individual factor, genetic diversity in populations, and the presence of comorbid disorders. To understand the genetic components of GTS etiopathology, we conducted an extensive review of the literature, compiling the candidate susceptibility genes identified through various genetic approaches. Even though several strong candidate genes have hitherto been identified, none of these have turned out to be major susceptibility genes yet.

## 1. Introduction

Gilles de la Tourette syndrome (GTS) is a complex early-onset neurodevelopmental/-psychiatric disorder with a prevalence between 0.52% and 0.77% in children [1,2]. GTS is one of the four tic disorders included in the fifth edition of the American Diagnostic and Statistical Manual of Mental Disorders (DSM-5): GTS, persistent (chronic) tic disorder (CTD), provisional tic disorder, and other specified and unspecified tic disorders. Diagnostic criteria for GTS are the presence of multiple motor tics and at least one vocal tic persisting for at least one year with onset of the first tic before age 18 [3]. Diagnostic criteria for CTD includes all of these criteria that exception that an individual has either motor or vocal tics, but not both. Provisional tic disorder meets all of the criteria for GTS except for the fact that tics do not last longer than 12 months. Finally, it is categorized as other specified and unspecified tic disorder when none of the above criteria are met [3]. Recently, the term “tic spectrum disorders” was suggested to replace the GTS and CTD diagnoses [4]. In addition to tics, most GTS individuals present a variety of symptoms due to comorbidities, the most common of which are obsessive-compulsive disorder (OCD) and/or attention deficit hyperactivity disorder (ADHD), and to a lesser extent autism spectrum disorder (ASD), migraine, anxiety, depression, sleep disorders, and rage attacks, implying an overlapping etiology [5,6,7,8,9,10]. Several neurotransmitter systems have been implicated in disease pathogenesis and amongst these, the dopaminergic and the serotonergic pathways are the most widely studied.

GTS is a complex disorder, where several environmental factors (such as streptococcus infections, birth complications and maternal smoking) are thought to interact with multiple genes in yet undiscovered ways [11]. Altered immune regulation is also suggested to predispose to inflammation and infection, thereby triggering GTS [11], and there is growing evidence for dysregulation of the brain’s resident immune cells, the microglia [12]. Several lines of evidence suggest that GTS has a strong genetic component [13], and it has been suggested as one of the non-Mendelian neuropsychiatric disorders with the highest heritability [9]. However, the identification of susceptibility genes has been challenging, which is likely due to the complex and heterogeneous genetic architecture of GTS, wherein common and rare variants in several different genes and biological pathways are involved. Even though neurophysiology and neuroimaging studies suggest that GTS is associated with altered synaptic neurotransmission, the pathophysiology is far from understood, which also hampers the identification of possible associated genetic factors. The overall results of the hitherto genetic studies suggest that in lieu of one or more major susceptibility loci, several rare and low-penetrance variants account for the genetic risk factors in GTS etiology.

We have performed an extensive review of the literature to compile the candidate GTS susceptibility genes identified through a candidate gene approach or data-driven studies such as GWAS or linkage analyses (Appendix A). In this review, we will highlight some of the most recent and notable studies in tic genetics, genes of historical significance in tic research, and studies focusing on candidate biological systems, such as the dopaminergic and serotonergic neurotransmission pathways.

## 2. Candidate Gene and Pathway Studies

Most of the studies utilizing a candidate gene/pathway approach have focused on the genes encoding the proteins of the neurotransmitter pathways of the cortico–basal ganglia–thalamo–cortical (CBGTC) loops. The CBGTC loops are the primary neural circuits associated with GTS pathogenesis, and they are altered in both structure and activity in GTS individuals [14,15,16]. The CBGTC loops connect specific regions of the cerebral cortex, the thalamus, and the basal ganglia, which is associated with several movement disorders [17]; and notably, GTS is also categorized as a movement disorder. Multiple neurotransmission systems present within the CBGTC loops, including dopaminergic, serotonergic, glutamatergic, and **γ**-aminobutyric acid (GABA)ergic systems, have been the target for GTS candidate gene studies.

### 2.1. The Dopaminergic Pathway

The most extensively studied neurotransmission system in GTS is the dopaminergic pathway (Figure 1). Upon the excitation of the dopaminergic neurons, dopamine is released from the synaptic vesicles of the presynaptic neurons into the synaptic cleft, where it is either transported back to the cytosol facilitated by the dopamine transporter DAT1 (encoded by *SLC6A3*) or interacts with post- or presynaptic dopamine receptors [18]. Dopamine receptors are categorized in two classes; D1-like receptors (D1 and D5 are encoded by *DRD1* and *DRD5,* respectively) and D2-like receptors (D2, D3, and D4 are encoded by *DRD2, DRD3,* and *DRD4,* respectively) coupled to excitatory and inhibitory G proteins, respectively [19]. Other players of the dopaminergic pathway include monoamine oxidase A (encoded by *MAOA*) and catechol O-methyltransferase (encoded by *COMT*), which metabolize unbound cytosolic dopamine, and dopamine β-hydroxylase (encoded by *DBH*), which catalyzes the conversion of dopamine to norepinephrine [18].

Aberrant regulation of dopamine uptake and release [20,21] and increased levels of dopaminergic innervation of the striatum (a component of the basal ganglia) have been reported in GTS individuals [22], and dopamine receptor blockers have successfully been used for the treatment of tics in GTS [21]. Dopamine receptors are also common drug targets in the treatment of other neuropsychiatric disorders such as schizophrenia and Parkinson’s disease and, to a lesser extent, ADHD [19].

Genes encoding the proteins of the dopaminergic system have thus been investigated extensively as candidate GTS susceptibility genes. The first association and linkage studies were conducted at the end of the previous century, where Comings et al. showed association between GTS and the *DRD2*, *DRD3*, *SLC6A3*, and *DBH* polymorphisms [23,24,25]. Subsequently, several other studies investigated the involvement of the dopaminergic pathway genes in GTS etiology [26,27,28,29]. In 2004, Díaz-Anzaldúa et al. [30] investigated 110 French Canadian trios and provided support for *DRD4* and *MAOA* as GTS susceptibility genes, but not for *DRD2, DRD3*, or *SLC6A3*. However, these results could not be replicated in other studies [31,32,33], with the exception of a single publication suggesting the involvement of *DRD4* in GTS etiology, particularly in Asian populations [34].

Later studies, however, reported an association between GTS and *DRD2*, especially the downstream Taq1A polymorphism (rs1800497): A meta-analysis including a total of 523 GTS individuals and 564 controls together with 87 GTS probands and 152 relatives from five studies concluded that Taq1A polymorphism might confer an increased risk of developing GTS, particularly in Caucasian populations [35]. The Taq1A polymorphism was later shown to be located within the coding region of the previously unknown *ANKK1* gene (the locus now called *ANKK1/DRD2*) which encodes a receptor interacting protein kinase. *ANKK1* may also be associated with the dopaminergic system in the brain, as *ANKK1* expression is affected by activation of dopamine receptors [36,37], but additional functional studies are needed to determine its role in GTS pathogenesis.

In a more recent study including 51 GTS individuals and 51 controls, the hypermethylation of *DRD2* was correlated to GTS and tic severity, and significant hypomethylation of *SLC6A3* was observed in GTS individuals with severe symptoms [38]. These results suggest that the altered epigenetic regulation of dopamine-related genes may play a role in GTS etiology, but this finding should be replicated in larger cohorts.

In conclusion, the above-mentioned studies have been conducted in small cohorts with limited statistical power and an unequivocal conclusion regarding the involvement of the dopaminergic pathway genes in GTS etiology cannot be drawn. Similarly, none of the later GTS GWASs identified any significant association, altogether suggesting that the genes of the dopaminergic pathway do not markedly contribute to GTS pathogenesis.

### 2.2. The Serotonergic Pathway

The serotonergic pathway, which is closely linked to the dopaminergic pathway, has also been studied in GTS, especially in OCD etiology [39]. Similar to dopamine, serotonin (5-hydroxytryptamine, 5-HT) is released from a presynaptic neuron into the synaptic cleft and is transported back to the cytosol by the serotonin transporter SERT (encoded by *SLC6A4*) or is bound to serotonin receptors. There are at least 14 different classes of serotonin receptors on the post- or presynaptic membranes [40]. Serotonin metabolism involves tryptophan 5-hydroxylase 2 (encoded by *TPH2*) which is responsible for the biosynthesis of serotonin from L-tryptophan; as for dopamine, MAOA is responsible for the degradation of serotonin to 5-hydroxyindole acetic acid (5-HIAA).

Interest in the serotonergic pathway in GTS etiology began when decreased levels of serotonin and 5-HIAA were detected in the cerebrospinal fluid and subcortical brain regions of GTS individuals [41,42,43], although these results could not be replicated in a later study [44]. In addition, serotonin reuptake inhibitors were also found to have an, albeit modest, effect on tic treatment [45,46].

Several serotonin receptors, including 5-HT1A, 5-HT1B, 5-HT2A, 5-HT2C, 5-HT3, and 5-HT4, are capable of facilitating and inhibiting dopamine activity in addition to that of serotonin [47,48,49], just as SERT can act both as a serotonin and dopamine transporter [50]. This has collectively led to the speculation that these two neurotransmitter pathways might be involved in various neuropsychiatric disorders in a combined manner. However, while the genes encoding the proteins of the dopaminergic pathway have been extensively studied in GTS, similar studies for the serotonergic pathway are scarce.

In 2010, common polymorphisms in several serotonin receptors (*HTR1A*, *HTR2A*, and *HTR2C*) were investigated in 87 Caucasian GTS individuals and 311 controls, and only the *HTR2C* polymorphisms Cys759Thr and Gly697Cys showed a nominally significant association with GTS [51]. These results could not be replicated in a Han Chinese cohort of 110 GTS individuals and 440 controls [52]. Similarly, polymorphisms of the serotonin receptors *HTR3A* and *HTR3B* could not be linked to GTS [53].

*SLC6A4* encoding SERT has repeatedly been investigated in GTS cohorts focusing particularly on the SERT-linked polymorphic region (5-HTTLPR) in the promoter region of *SLC6A4*. The 5-HTTLPR is a 43bp insertion/deletion giving rise to a long (L) allele and a short (S) allele (Figure 2). The L_AC_ haplotype, which is a combination of the L allele and the major alleles of the rs25531 (A/G) and rs25532 (C/T) SNPs and which is associated with the highest rate of *SLC6A4* mRNA expression (Figure 2), was found to be more prevalent in 151 GTS individuals compared to 858 controls [54]. Recently, *SLC6A4* promoter variants and mRNA levels were investigated in a Danish cohort comprising 72 GTS individuals and 87 controls. *SLC6A4* mRNA expression levels were found to be higher in GTS individuals, particularly when only considering the individuals with the L_AC_/L_AC_ genotype [55]. In earlier studies, an association between GTS and the 5-HTTLPR variants was not detected [51,56,57]. These contradictory results may be due to the previously unnoticed effect of the rs25531 (A/G) and rs25532 (C/T) SNPs on *SLC6A4* mRNA expression, as it is likely that higher expression of *SLC6A4* mRNA is associated with the 5-HTTLPR/rs25531 L_A_ allele but not with the L_G_ and S alleles [58] (Figure 2). Most recently, the exome sequencing of 13 multiplex GTS families suggested *SLC6A4* as a high confidence GTS risk gene [59].

*TPH2* has also been the focus of several candidate gene studies. The C/C genotype and the C allele of rs4565946 (C/T) were associated with GTS in a German cohort comprising 98 GTS individuals and 178 ancestry-matched controls [60]. In a Han Chinese cohort comprising 149 children with tic disorders and 125 controls, the T allele was linked to tic disorders suggesting an ancestry-dependent association [61]. Subsequently, a large association study including 412 North American, European, and South Korean GTS/CTD trios and quartets and a replication cohort consisting of 1285 GTS individuals and 4964 controls found a nominally significant association of rs4565946 to GTS/CTD, though it ceased to be significant after correction for multiple testing [62].

The data regarding the role of the serotonergic pathway in GTS etiology are limited and again, individual studies have been of limited statistical power due to their low sample size. However, when studies of the dopaminergic system are taken into consideration, it is plausible to hypothesize that the serotonergic system may play a complex role in GTS etiology that is directly and/or indirectly in combination with or through modulating dopaminergic neurotransmission. Additional neurotransmitter pathways (e.g., glutamatergic and GABAergic) may also be involved in the pathology of GTS, as the dysfunction of one pathway may affect others due to interaction or self-regulation among them.

## 3. Linkage Analyses, GWAS & Other Studies

Besides the hypothesis-driven approach, several data-driven genetic studies such as linkage analyses and GWASs have been conducted to elucidate the underlying genetic risk factors for GTS. In this review, we will highlight *SLITRK1* and *IMMPL2* due to their historical significance, as well as candidate susceptibility genes such as *HDC, CELSR3, FN1, ASH1L*, *FLT3, NRXN1*, and *CNTN6,* which have been identified in independent studies. All of the other studies are described very briefly in Appendix A.

### 3.1. SLITRK1

*SLITRK1* (Slit and Trk-like 1) is the most investigated GTS candidate gene, since one of the breakpoints of a de novo inversion of chromosome 13 in a child with GTS was mapped within its proximity [63]. Functional studies showed that neurite outgrowth was affected by a truncating *SLITRK1* variant (varCDf) identified in a GTS individual, and a missense variant (var321) in the 3′ untranslated region (3′-UTR) of the gene altered a microRNA binding site leading to decreased mRNA expression [63].

However, several subsequent association studies, predominantly in European populations, could not detect an association between GTS and var321 or other *SLITRK1* variants [64,65,66,67,68]. A cohort study comprising 114 Costa Rican and 193 Ashkenazi Jewish GTS probands and their parents suggested population stratification rather than an association between var321 and GTS, as the variant was found in several of the Ashkenazi parents and one of the Ashkenazi controls [65]. A later study did not find any evidence for a var321 founder allele amongst Ashkenazi Jews but showed an association between var321 and GTS [69].

Other studies demonstrated an association between other *SLITRK1* variants and GTS individuals from European (n = 447 total), Canadian (n = 231 total), and East Asian populations (n = 252 total) [70,71,72,73,74,75]. A total of four of the variants (c.1061G > C, c.1158G > T, c.892C > T, and c.1252A > T) were concluded to be rare but deleterious variants [70].

The first two GTS GWASs were conducted on 1285 GTS individuals (4964 controls) and 2723 individuals with GTS and/or OCD (5667 controls) [76,77] and more recently, a GWAS meta-analysis with a larger sample size (4819 GTS individuals and 9488 controls) was conducted [78]. All of the cases and the controls of these three studies were of European ancestry, and none of the studies identified an association between GTS and *SLITRK1*. In the GWAS conducted by Scharf et al., the SNP rs7336083 residing in the intergenic region between *SLITRK1* and *SLITRK6* was one of the highest-ranking loci, but it did not reach the threshold of genome-wide significance [76]. However, in a cohort comprising 399 Han Chinese trios, an association between *SLITRK6* and GTS could not be detected [79]. Similarly, *SLITRK5*, which has been associated with OCD and is within the same cluster as *SLITRK1* and *SLITRK6*, could not be linked to GTS [80,81], although a de novo *SLITRK5* missense mutation was recently identified in a proband with GTS and ASD [82].

SLITRK1 promotes neurite outgrowth and *SLITRK1* is predominantly expressed in the brain, including in the CBGTC circuits [83]. In addition, the six proteins belonging to the SLITRK family play important roles in the development of the central nervous system and neuronal processes and have been implicated in several other neurodevelopmental and -psychiatric disorders [84]. The *SLITRK*-genes have therefore been attractive candidates in the search for the genes involved in GTS etiology. However, it is likely that these genes, especially *SLITRK1*, do not play a major role in GTS etiology.

### 3.2. IMMPL2

Similar to *SLITRK1*, *IMMPL2* (inner mitochondrial membrane peptidase, subunit 2) was suggested as a candidate GTS gene by the finding of a chromosome aberration in a GTS individual, where the gene was disrupted by a 7q31 breakpoint [85], a region previously linked to GTS [86]. The role of *IMMPL2* and the 7q31 region in GTS etiology was further supported by the finding of another Caucasian individual with tics and a disruption of *IMMP2L* [87] and a family based association study (86 French Canadian GTS trios) suggesting an association between GTS and several 7q31 markers, including a tendency of association with an intragenic *IMMP2L* marker [88].

Even though deleterious *IMMPL2* variants could not be detected in 39 European GTS individuals [89], a chromosome microarray study of a Danish cohort (188 GTS individuals and 316 controls) showed a significantly increased number of intragenic *IMMPL2* deletions in GTS individuals, and some of these deletions affected an alternatively spliced isoform [90]. As IMMP2L functions in the mitochondria and as mitochondrial dysfunction is linked to movement disorders, fibroblasts from the Danish GTS individuals and controls, both with or without deletions, were assessed on various parameters relating to mitochondrial dysfunction, without detecting a significant disruption in mitochondrial function [91]. Finally, a recent meta-analysis of previously reported GTS candidate genes in European GTS individuals and controls detected an association between GTS and *IMMPL2* [92], further implicating *IMMPL2* in GTS etiology. However, how a defective IMMP2L function may be involved in GTS pathogenesis is still unknown.

### 3.3. HDC

*HDC* encodes the histidine decarboxylase enzyme essential for histamine synthesis and hence for histaminergic neurotransmission, and the striatum has a high density of histamine receptors which modulate both dopaminergic and serotonergic neurotransmission [93]. *HDC* was first implicated as a candidate GTS susceptibility gene by the identification of a nonsense truncating variant (Trp317Ter) segregating with the disorder in a Caucasian, nonconsanguineous two-generation family (eight siblings and their parents) with an extremely high prevalence of GTS, as only a single family member was unaffected [94]. This variant was not present in 3000 Caucasian controls, in a replication cohort of 720 GTS individuals, nor in 360 controls not screened for psychiatric disorders. These results suggested the Trp317Ter variant is a very rare genetic GTS risk factor with high penetrance. This study implicated disrupted histamine production in GTS etiology and raised the histaminergic hypothesis. Since then, several studies have found an association between GTS and *HDC* variants in populations of European ancestry [70,72,95], and *HDC* knockout mice exhibited tic-like stereotypies [96]. A genome-wide analysis of de novo copy number variations (CNVs) in 460 Caucasian GTS individuals (including 148 trios) and 1131 ancestry-matched controls showed enrichment of genes within the histamine receptor (H1R and H2R) signaling pathways in GTS individuals, further supporting the histaminergic hypothesis [97]. However, two studies among Han Chinese GTS individuals, one with a case-control approach (120 cases and 240 controls) [98] and the other with a family-based approach (241 nuclear family trios) [99], did not detect an association between *HDC* and GTS. Taken together, this could indicate that a causal role for *HDC* in GTS etiology primarily applies to individuals of European ancestry, in which case it has the potential to be an important candidate gene with high penetrance in very rare cases, an uncommon feature in complex disorders such as GTS.

### 3.4. CELSR3, WWC1, FN1, and NIPBL

In 2017, exome sequencing of 511 GTS trios with various ancestries suggested that de novo damaging variants in approximately 400 genes contributed to the genetic risk load in 12% of the individuals [100]. In addition, four likely susceptibility genes with multiple de novo damaging variants were identified in unrelated GTS individuals: *CELSR3* (cadherin EGF LAG seven-pass G-type receptor 3), *WWC1* (WW and C2 domain containing 1), *FN1* (Fibronectin 1), and *NIPBL* (Nipped-B-like) [100]. In a follow-up study, additional variants of these four genes were identified [101]. All of these variants were either absent or present in very low frequency in the gnomAD database, which comprised about 250,000 individuals [102]. The two former genes were suggested as high confidence GTS risk genes and the two latter were suggested as probable GTS risk genes. *FN1*, which encodes a cell adhesion protein, was further implicated in GTS etiology by the finding of a de novo missense variant in a GTS/ASD proband [82]. Similarly, variants of *CELSR3*, which encodes a non-classic cadherin transmembrane receptor, were identified in two European [72] and one Chinese GTS individual [103].

### 3.5. ASH1L

One of the most recent and quite notable additions to the list of potential GTS candidate genes is *ASH1L* (ASH1-like histone lysine methyltransferase), identified through the exome sequencing of 100 GTS trios [104]. *ASH1L* was suggested to be associated with GTS based on a transmission disequilibrium test (TDT) as well as a finding of de novo variants. Of the nineteen reported damaging *ASH1L* variants, five were present in very low frequency in the gnomAD database. The association was replicated through targeted sequencing of 524 GTS samples (and 2822 East Asian ExAC controls). Some of the variants altered the methyltransferase activity of the protein, and *Ash1l+/−* transgenic mice manifested compulsive and tic-like behaviors that could be rescued by a tic-relieving drug. The disruption of *Ash1l* also affected dopaminergic modulation in the dorsal striatum in the basal ganglia [104], altogether indicating a role of *ASH1L* in GTS physiopathology.

### 3.6. FLT3

Recently, in a GWAS with a cohort comprising 4819 GTS individuals and 9488 controls, Yu et al. reported a single genome-wide significant SNP (rs2504235) within *FLT3*, encoding FMS-like tyrosine kinase 3 [78]. rs2504235 is in strong linkage disequilibrium with the common *FLT3* missense variant rs1933437 (p.Thr227Met), which in the same study had a p-value of 8.2 × 10^−8^ [78]. This latter variant drove the association between GTS and a lymphocytic gene set in a recent large pathway analysis based on genome-wide genotypic data from 3581 GTS individuals and 7682 controls [105]. The association between lymphocytic genes, *FLT3*, and GTS not only provides further support for a role of *FLT3* in GTS, but also suggests an involvement of a neuroinflammatory element in disease pathogenesis. In addition, the pathway analysis also implicated ligand-gated ion channel signaling and cell adhesion and transsynaptic signaling processes [105].

### 3.7. NRXN1 and CNTN6

Rare CNVs are known to be associated with a number of neuropsychiatric disorders [106], including GTS [97]. The investigation of rare CNVs in a large cohort (2434 GTS individuals and 4093 controls) of European ancestry showed an increased global burden of CNVs in the GTS sample [107]. Notably, *NRXN1* deletions and *CNTN6* duplications were present in 1% of the GTS individuals and survived genome-wide correction for multiple testing and were suggested as definitive GTS susceptibility loci. *CNTN6* encodes a cell adhesion molecule, contactin-6, and has been implicated in ASD and intellectual disability [108,109]. *NRXN1* encodes the presynaptic cell adhesion molecule neurexin 1 which is involved in glutamatergic and GABAergic neurotransmission and synaptogenesis [110]. *NRXN1* deletions have previously been associated with GTS, although the sample sizes have been considerably smaller (111 Caucasian GTS individuals with 73 ancestry-matched controls and 263 Latin American GTS individuals with 285 ancestry-matched controls, respectively) [111,112]. Furthermore, *NRXN1* deletions have been implicated in other neurodevelopmental and -psychiatric disorders, including ASD and schizophrenia [113,114,115]. The two susceptibility loci are important additions to the list of candidate GTS susceptibility loci and demonstrate the importance of rare structural variations in GTS etiology.

## 4. Discussion—Future Opportunities and Challenges

GTS is a disorder with complex and heterogeneous genetic architecture, seemingly due to a combination of rare genetic risk factors, either de novo or inherited, and an inter-individual variation in polygenic burden. Several lines of evidence point towards neurotransmitter mediated involvement of the CBGTC circuits in GTS pathology. We hypothesize that a link between the dopaminergic and serotonergic pathways may involve MAOA, which is responsible for the degradation of both cytosolic dopamine and serotonin in presynaptic neurons, and potential feedback mechanisms regulating MAOA levels or activity. If this is the case, an overexpression of a neurotransmitter transporter such as SERT as reported by Hildonen et al. [55] might result in an upregulation of MAOA due to increased levels of cytosolic serotonin. This could potentially have a more global effect, leading to increased degradation of both serotonin and dopamine.

To identify GTS susceptibility genes within neurotransmission pathways, early studies applied the candidate gene approach in relatively small cohorts, investigating a few polymorphisms at a time. Today, technological advances allow for larger and hypothesis-free studies such as GWAS, exome sequencing, and whole genome sequencing (WGS). The rationale behind GWAS is the common disease-common variant hypothesis and requires very large cohorts when studying highly complex diseases, where rare variants often play a more considerable role. The first GWAS in GTS was conductedin a relatively small cohort, which likely explains why none of the individual SNP signals reached a genome-wide level of significance [76]. The more recent GWAS conductedin a cohort that was nearly four times larger, identified a single SNP within *FLT3* reaching genome-wide significance [78], consistent with the call for large sample sizes in these studies. Due to the fall in the running costs, exome and genome sequencing are now attractive alternative methods for detecting novel and rare variants, and exome sequencing in particular has been utilized in several recent GTS studies [100,101,104,116,117]. Through pathway analyses, Tsetsos et al. identified three significant gene sets that may be involved in GTS pathogenesis, demonstrating yet another useful approach in the search for candidate GTS susceptibility genes [105]. Meta-analyses and replication studies of the most significant hits from all of these screening studies, in combination with functional analyses such as investigation of mRNA expression, methylation patterns or the more cumbersome cellular studies, would be future approaches for a better understanding of the underlying genetic risk factors in GTS.

Genetic variation due to ancestry and clinical heterogeneity, including the presence of comorbidities, are confounding factors hampering the identification of susceptibility genes in complex disorders. GTS has been reported to occur less frequently among individuals of African descent compared to Caucasian Europeans [118], although it is not clear whether this is due to decreased awareness of the disorder in specific populations. As accounted for throughout this review, there also appears to be some general distribution differences of genetic susceptibility factors between populations of different ancestry. This potential confounding effect of a hidden population substructure further challenges approaches searching for common, and more notably, rare variants that are associated with the disorder. Furthermore, there is a risk of stratification on genetic ancestry, particularly in GWAS [119], which should be taken into consideration in the search for potential candidate GTS genes.

The presence of comorbidities likewise impedes the identification of GTS susceptibility genes, comparison between studies, and meta-analyses introducing additional levels of heterogeneity. In several GTS studies, CTD and other tic disorders are also included and the presence of comorbidities, occasionally apart from OCD, are not taken into consideration. Furthermore, in some studies, not all GTS individuals are diagnosed using clinically validated instruments, and some are noted as self-reported. Diagnosis of GTS is further challenged by the suggestion that there may be more than one GTS phenotype [120]. Cross-disorder studies have suggested a significant proportion of GTS heritability to be shared with OCD, ADHD, major depression disorder, and migraine [121,122], and heritable cross-disorder endophenotypes of GTS and OCD have been identified [123]. A recent cross-disorder study reported 11 previously unidentified regions likely associated with GTS, ADHD, and ASD as well as two separate pleiotropic regions associated with both GTS and OCD, indicating that the genetic etiology shared between GTS and OCD might differ from that shared between GTS, ADHD and ASD [124]. On the other hand, another cross-disorder GWAS investigating the genetic architecture behind OCD and GTS suggested that the susceptibility genes behind OCD with comorbid GTS might differ from those involved in OCD alone [77]. Similarly, individuals with pure GTS (without comorbidities) were shown to have no family history of OCD [125], altogether indicating the presence of distinct genetic components of GTS.

## 5. Conclusions

The collaborative research conducted over the last few years and the investigation of large cohorts to identify the genetic susceptibility factors involved in GTS etiology are beginning to bear fruit. The employment of uniform methodology and clinically homogeneous cohorts taking comorbidities and ancestry into consideration are crucial in achieving more consistent and comparable results in future studies. Furthermore, functional analyses and cross-disorder studies might offer novel insights into the etiology and pathophysiology of GTS and overlapping neurodevelopmental, -psychiatric, and movement disorders. A better understanding of the complex etiology of GTS and new insights into the pathophysiology of GTS, which is far from being fully understood, are crucial in new treatment strategies for this disorder.

## Figures and Tables

**Figure 1 genes-12-01321-f001:**
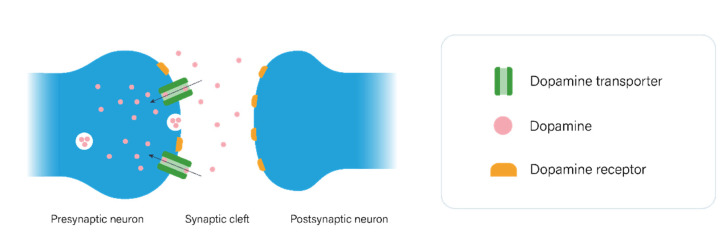
Visualization of dopaminergic neurotransmission in the synaptic cleft. Dopamine is released from the vesicles in the presynaptic neuron into the synaptic cleft, where it binds to either presynaptic or postsynaptic dopamine receptors or is transported back to the cytosol of the presynaptic neuron by dopamine transporters.

**Figure 2 genes-12-01321-f002:**
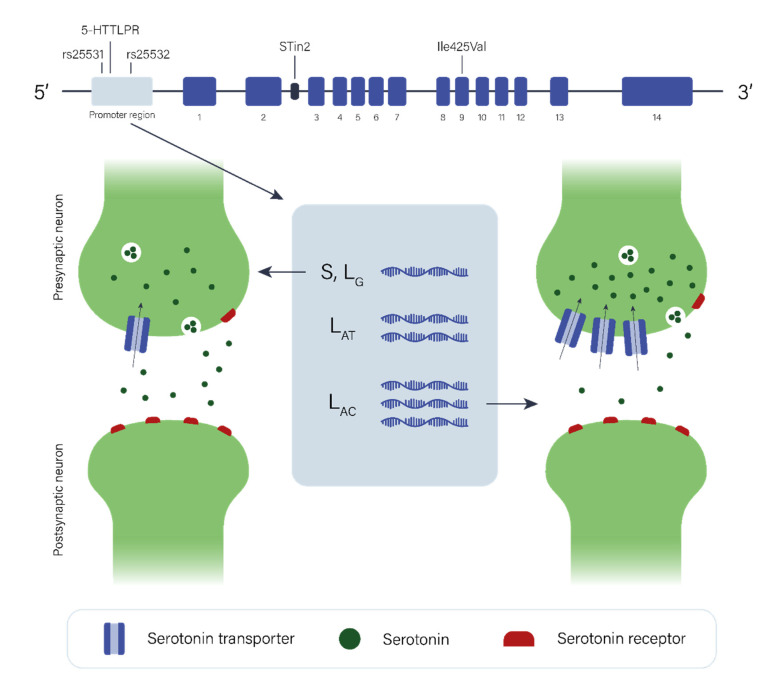
*SLC6A4* and serotonergic neurotransmission. *SLC6A4* gene, exon numbers, and the polymorphisms 5-HTTLPR, rs25531, rs25532, STin2, and Ile425Val in the top part of the figure. The different 5-HTTLPR/rs25531/rs25532 three-locus haplotypes (**bottom middle**) are likely to affect *SLC6A4* mRNA expression (**bottom middle**). It is hypothesized that S and L_G_ haplotypes result in low *SLC6A4* mRNA expression (**bottom left**), while the L_AC_ haplotype results in *SLC6A4* overexpression, leading to increased SERT in the presynaptic neuron followed by increased serotonin clearance from the synaptic cleft (**bottom right**). 5-HTTLPR: SERT-linked polymorphic region; S: short allele of 5-HTTLPR; L_G_: long allele of 5-HTTLPR and minor allele of rs25531; L_AT_ and L_AC_: 5-HTTLPR/rs25531/rs25532 three-locus haplotypes, differing in rs25532 alleles.

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
