# Peer review of "Candidate Genes and Pathways Associated with Gilles de la Tourette Syndrome—Where Are We?"

_genes, 2021, doi:10.3390/genes12091321_

Round 1

Reviewer 1 Report

The authors provide an extensive review of the existing literature concerning genetic factors associated with Gilles de la Tourette syndrome. The manuscript is highly relevant and I do appreciate the idea to combine data according to the biological pathway influenced by the different genes.

In general, the manuscript is interesting and well written.

I have a however a few points that I suggest to improve:

1) The last two sentences of the abstract are not informative and should be replaced with something else to help the reader understand what he will find in the paper.

2) The etiopathogenesis of Gilles de la Tourette Syndrome (and of tic disorders in general) is far from being fully understood. The authors might point out more in a more explicit way that this is a problem also for research of possibile associated genetic factors.

3) Among the comorbidities of Gilles de la Tourette Syndrome, I can't find anxiety which is however a frequently reported comorbiditi (see e.g. https://pubmed.ncbi.nlm.nih.gov/32519083/).

4) The role of non genetic factors should at least be mentioned, despite not being the focus of the paper.

Author Response

1) The last two sentences of the abstract are not informative and should be replaced with something else to help the reader understand what he will find in the paper.

We have replaced the last two sentences.

2) The etiopathogenesis of Gilles de la Tourette Syndrome (and of tic disorders in general) is far from being fully understood. The authors might point out more in a more explicit way that this is a problem also for research of possible associated genetic factors.

We underlined this point in the introduction and conclusion.

3) Among the comorbidities of Gilles de la Tourette Syndrome, I can't find anxiety which is however a frequently reported comorbiditi (see e.g. https://pubmed.ncbi.nlm.nih.gov/32519083/).

Thank you for bringing this to our attention. The text has been updated and anxiety included amongst the comorbidities.

4) The role of non genetic factors should at least be mentioned, despite not being the focus of the paper.

Non-genetic factors are now included in the introduction.

Reviewer 2 Report

Gilles de la Tourette's syndrome, named after a nineteenth-century French neurologist, is a tic disorder that is characterized by childhood onset, duration of more than a year, motor tics, vocal tics, and a waxing and waning of symptoms. Obsessive-compulsive disorder and attention-deficit/hyperactivity disorder occur in many patients. Most cases appear to be hereditary, but a specific inheritance pattern is uncertain. Tourette's syndrome may be related to the overactivity of the dopaminergic systems. I have  following concerns to improve the existing version:    

-The first line of both abstract and introduction are almost same. 

-Chapter 2. Candidate Gene and Pathway Studies is interesting however, Fig 1 is should be updated. 

-Line 101, cite reference just after Díaz-Anzaldúa et al. 

-Authors said “Later studies, however, reported an association between GTS and  DRD2, especially   

the downstream Taq1A polymorphism (rs1800497)” Give citation for this info. 

-How ANKK1 is linked to the dopaminergic system? 

-Line 141, add the name of serotonin receptors? 

-In some important and conflicting clinical studies add significant values. 

-The quality of English should be improved. 

-A comparative table might be effective for readers for Linkage Analyses, GWAS & Other Studies. 

-Add a section regarding future opportunities and challenges. 

-Write the conclusion separately. 

Author Response

The first line of both abstract and introduction are almost same.

The text has been modified.

Chapter 2. Candidate Gene and Pathway Studies is interesting however, Fig 1 is should be updated.

Thank you. The figure and figure text has been updated.

Line 101, cite reference just after Díaz-Anzaldúa et al.

The text has been updated as suggested.

Authors said “Later studies, however, reported an association between GTS and DRD2, especially the downstream Taq1A polymorphism (rs1800497)” Give citation for this info.

The citation is the following meta-analysis. The text has been updated with a colon to make it clearer.

How ANKK1 is linked to the dopaminergic system?

More information is added in the text.

Line 141, add the name of serotonin receptors?

The specific serotonin receptors have been listed.

In some important and conflicting clinical studies add significant values.

We have not fully understood this comment as clinical studies were not focus of the review.

The quality of English should be improved.

The authors have improved the text and a native English speaker went through the manuscript.

A comparative table might be effective for readers for Linkage Analyses, GWAS & Other Studies.

In the supplementary table all the different studies carried out are listed. We think that an additional table comparing different studies will be redundant and comparison of different studies was not the main focus of the review.  

Add a section regarding future opportunities and challenges.

Future opportunities and challenges are already mentioned in the discussion section, therefore we changed the name of this section covering its content.

Write the conclusion separately.

The conclusion is now separated from the discussion and expanded.

Round 2

Reviewer 2 Report

The paper can be accepted for publication.